# Kānuka Trees Facilitate Pasture Production Increases in New Zealand Hill Country

Thomas H. Mackay-Smith *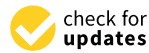, Ignacio F. López, Lucy L. Burkitt and Janet I. Reid

School of Agriculture and Environment, Massey University, Palmerston North 4410, New Zealand; i.f.lopez@massey.ac.nz (I.F.L.); l.burkitt@massey.ac.nz (L.L.B.); j.i.reid@massey.ac.nz (J.I.R.)
* Correspondence: t.mackaysmith@massey.ac.nz

**Abstract:** 'Tree-pasture' silvopastoral systems have the potential to become transformative multifunctional landscapes that add both environmental and economic value to pastoral farms. Nevertheless, no published study has found increased pasture production under mature silvopastoral trees in New Zealand hill country. This study takes a novel approach to silvopastoral research in New Zealand, and investigates a genus that has similar bio-physical attributes to other global silvopastoral trees that have been shown to increase pasture production under their canopies, with the aim of finding a silvopastoral genera that can increase pasture production under tree canopies compared to open pasture in New Zealand. This study measures pasture and soil variables in two pasture positions: under individually spaced native kānuka (*Kunzea* spp.) trees (kānuka pasture) and paired open pasture positions at least 15 m from tree trunks (open pasture) at two sites over two years. There was 107.9% more pasture production in kānuka pasture positions. The soil variables that were significantly greater in kānuka pasture were Olsen-P (+115.7%, $p < 0.001$), K (+100%, $p < 0.001$), Mg (+33.33%, $p < 0.01$), Na (+200%, $p < 0.001$) and porosity (+8.8%, $p < 0.05$), and Olsen-P, porosity and K best explained the variation between kānuka pasture and open pasture positions. Volumetric soil moisture was statistically similar in kānuka pasture and open pasture positions. These results are evidence of nutrient transfer by livestock to the tree-pasture environment. Furthermore, as there was a significantly greater porosity and 48.6% more organic matter under the trees, there were likely other processes also contributing to the difference between tree and open pasture environments, such as litterfall. These results show that kānuka has potential to increase pasture production in New Zealand hill country farms and create multifunctional landscapes enhancing both production and environmental outcomes in pastoral farms.

**Keywords:** agroforestry; treeless pasture; *dehesa*; ñire; oak; poplar; ecosystem services



## 1. Introduction

In many situations, 'tree-pasture' silvopastoral trees can become 'islands of fertility' [1–3], build soil organic matter [3–5], conserve soil moisture [6,7] and improve soil structure [8,9]. Moreover, silvopastoral systems can be carbon sinks in terms of above and below ground biomass and potential increases to soil carbon [10,11], improve the local agricultural microclimate [5,12], provide shelter and shade to livestock [13–15] and provide habitats to local bird populations [16–18].

One region that could benefit from wider use of silvopastoralism is New Zealand hill country, which is an agricultural area that is defined as steep or hilly land (>15°), having an altitude <1000 m asl and pastoral farming as its main land use (sheep, cattle and deer) [19]. Silvopastoralism is already used in New Zealand hill country to mitigate soil erosion [20–23]. The most commonly planted and researched silvopastoral genus is poplar (*Populus* spp.) [23–26]. Poplars have been selected as soil conservation trees because of their quick growth [27,28], large root system [22,29,30], high evapotranspiration rates when growing [23,31,32] and ability to plant the trees as unrooted sharped coppiced poles

in the presence of grazing sheep and cattle [30]. Nevertheless, past published studies only report negative impacts to pasture production [24], with reductions ranging from 12 to 65% compared to open pasture [24,33–37].

There are, however, many examples of research reporting increased pasture production under silvopastoral trees in other systems [38–40]. In three Spanish *dehesa* silvopastoral sites with annual rainfall ranging from 452 mm to 661 mm, mean pasture production increased by 19% under holm oaks (*Quercus ilex* L.) compared to open pasture [40]. Frost and McDougald [39] found on average 63% and 50% more pasture production under the canopy of blue (*Q. douglasii* Hook. & Arn.) and interior live (*Q. wislizeni* A. DC.) oak compared to open pasture, respectively, at sites with an average rainfall of 487 mm. This effect has also been shown in the ñire (*Nothofagus antarctica* (G. Forst.) Oerst.) forests of southern Patagonia, Argentina, with ~700 mm annual rainfall, with maximum pasture production being achieved at a 50–60% canopy cover compared to open pasture in a severely water stressed site [7].

The tree attributes of poplars contrast with the southern European and Californian oak silvopastoral systems, and the ñire forests of southern Patagonia. For instance, there is evidence that these other silvopastoral trees not in New Zealand grow slower than poplars (although a study of all these trees growing under the same conditions has not been performed) [23,30,41–43], and so use less water and nutrients during establishment [44]. Moreover, the height of the main trees used in southern Europe (*Q. ilex* and *Q. suber* L.) and in Patagonia (*N. antarctica*) are typically between 4 m and 15 m [42,45–47], and those in California (*Q. douglasii* and *Q. wislizeni*) are typically between 7 and 20 m in height [48]. This compares to poplars that are >30 m [49], and could pose a challenge for pasture production because larger trees have been shown to use more water [44,50,51].

Furthermore, the systems in southern Europe, California and southern Patagonia use native trees [42,52,53], which is most likely why the trees in the *dehesa* system in Spain can live up to 250 years [52], and those in the ñire forests of Patagonia can live up to at least 180 years [42]. This compares to poplars, which often experience wind damage and leaf rust, and the recommended management practice is to fell and replant the trees after 40 years [54]. Slower growing silvopastoral trees that facilitate an increase in pasture production, and are intergenerational, should be more advantageous in New Zealand tree-planting situations where the soil is not highly susceptible to soil erosion.

Kānuka (*Kunzea* spp.) is a native genus that has 10 endemic species in New Zealand [55]. Many of these species are trees that are naturally common in New Zealand hill country [55–57], which have similar attributes to *N. antarctica* and the oaks mentioned above. For instance, kānuka grow slower [28,30,43,58] and are smaller (8–20 m high) [49] than poplars, so most likely compete less for soil resources [44,50,51]. Moreover, kānuka is a native to hill country and is reported to grow for at least 300 years [58]. These similarities between kānuka and other native silvopastoral trees globally leads to the hypothesis that kānuka will increase the availability of water and nutrients for pasture growth because of reduced competition, resulting in intergenerational positive impacts to pasture production in New Zealand hill country.

This is the first time kānuka has been studied in terms of its impact on pasture production. If kānuka increases pasture production under its canopy, this may not only be transformative for New Zealand pastoral farming, but it will provide evidence as to whether there are universal tree attributes responsible for positive pastoral outcomes in silvopastoral systems. If these can be found, this will greatly help in the creation, design and management of future silvopastoral systems. The objectives of this study are to (1) evaluate the impact of kānuka on pasture production when compared to equivalent open pasture positions, and (2) identify and discriminate the variables that contribute to pasture production differences between tree and open pasture positions.

## 2. Methods

### 2.1. Study Areas

This study was undertaken at two sites in the North Island of New Zealand. The first was in the Wairarapa region, ~10 km north of Martinborough (Wairarapa site), and the second was in the Hawkes Bay region, ~20 km south of Waipukurau (Hawkes Bay site) (Table 1; Figure 1). Two sites with similar environmental and soil conditions were selected to increase the reliability of the results (Table 1).

**Table 1.** Site characteristics for the two study sites.

| Study Site | Wairarapa | Hawkes Bay |
|---|---|---|
| Location | 41°08′41.3″ S, 175°29′58.3″ E | 40°08′25.9″ S, 176°23′39.1″ E |
| Region | Wairarapa, North Island of New Zealand | Hawkes Bay, North Island of New Zealand |
| Elevation (m) | 122 | 288 |
| Basement rock | Sandstone | Mudstone |
| New Zealand soil classification | Mottled Argillic Pallic Soil [59] | Mottled Argillic Pallic Soil [59] |
| US soil classification | Ustalf [59] | Ustalf [59] |
| Topsoil type | Silt loam | Silt loam |
| Subsoil type (B horizon) | Silty clay loam | Silty clay loam |
| Mean 30-year annual rainfall (mm) | 903 (min: 548; max: 1297), station 2631, 6.6 km from the site, elevation: 61 m [60] | 883 (min: 527; max: 1483), station 2523, 5.8 km from the site, elevation: 153 m [60] |
| Mean 10-year annual temperature (°C) | 18.3 (min: 17.5; max: 19.0), station 21938, 15.0 km from the site, elevation: 22 m [60] | 16.7 (min: 15.8; max: 17.5), station 25820, 15.3 km from the site, elevation: 341 m [60] |
| Paddock topography | Moderately to severely steep (15–40°) | Rolling to moderately steep (10–35°) |
| Livestock operation | Sheep and beef | Sheep and beef |
| Aspect | NE | NW |
| Measurement position slope gradient | 20–25° | 20–25° |

Both sites were on typical commercial sheep and beef farms with naturalized and permanent pasture. The most common pasture species in hill country are browntop (*Agrostis capillaris* L.) and perennial ryegrass (*Lolium perenne* L.); however, the pasture species composition of hill country pastures varies depending on the environmental conditions [61,62]. Individual *Kunzea robusta* de Lange et Toelken (kānuka) trees grow throughout the paddocks at both study sites at ~10 trees ha$^{-1}$ to ~2000 trees ha$^{-1}$ (Figure 1B,C). It is likely that the trees at both sites established naturally as seedlings following land clearance for grazing.

At the Wairarapa site, livestock was rotationally grazed for 2 to 3 days at a time throughout the year with a grazing intensity of 40 lambs ha$^{-1}$ day$^{-1}$, 57 ewes ha$^{-1}$ day$^{-1}$, 9.1 Angus cows ha$^{-1}$ day$^{-1}$ and 3.4 Friesian bulls ha$^{-1}$ day$^{-1}$. At the Hawkes Bay site, pregnant ewes during lambing were set stocked for about 1 month in spring and summer at a stocking rate of 5.7 ewes ha$^{-1}$. The paddock was also rotationally grazed by Angus cows for one week at a time throughout the year at 0.8 cows ha$^{-1}$ day$^{-1}$. Fertilizers that contained 21.5 kg P ha$^{-1}$ and 37 kg S ha$^{-1}$ were annually surface-applied at the Wairarapa site in early summer. The annual fertilization at Hawkes Bay was either 25 kg N ha$^{-1}$ and 28.75 kg S ha$^{-1}$, or 25.8 kg P ha$^{-1}$ and 42 kg S ha$^{-1}$, which was surface-applied in winter. There were also some years at the Hawkes Bay site where no fertilizer was applied because of sufficiently high soil P fertility.

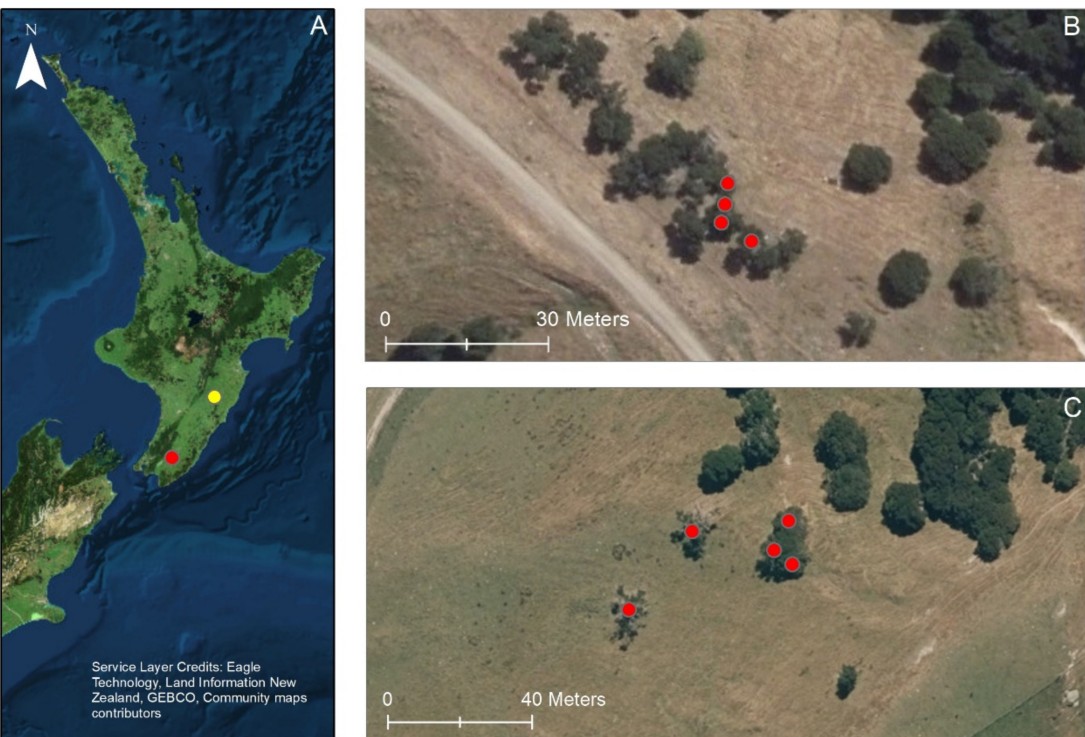

**Figure 1.** Study site locations and individual trees evaluated at each site. (**A**): Location of the study sites in New Zealand (the red dot is the Wairarapa site and the yellow dot is the Hawkes Bay site). (**B**): The studied kānuka trees at the Wairarapa site (red dots show the individual trees evaluated). (**C**): The studied kānuka trees at the Hawkes Bay site (red dots show the individual trees evaluated). (**B**,**C**) are from the same satellite layer as (**A**).

### 2.2. Study Design and Measurements

At each study site, pasture measurement positions were in pairs that represented the two treatments (kānuka pasture and open pasture). One 1.5 m × 1.5 m pasture position was half-way between the edge of the tree canopy and stem of individually spaced kānuka (kānuka pasture), and the other was in an equivalent pasture position (with a similar slope gradient and slope position) in open pasture at least 15 m from the nearest tree trunk (open pasture). Each open pasture position was at least 15 m from the trunk of the paired kānuka pasture position because the drip line (edge of canopy) was ~5 m from the trunk for all trees, and a distance of at least three times the drip line was selected as 'open pasture' [4]. Other authors have selected 2.5 times the drip line for open pasture positions [3,9], but 3 times was selected in the current study to maximize the contrast between open and kānuka pasture positions.

There were four pasture position pairs at the Wairarapa site and five at the Hawkes Bay site. This represented nine tree replicates in total for each treatment (kānuka pasture and open pasture) (Figure 1B,C). One tree replicate represented the experimental unit. The height of the trees was ~10 m at Wairarapa and between 10 m and 15 m at Hawkes Bay. The exact age of the trees are unknown, but historic aerial imagery show that the trees at the Wairarapa site are over 80 years old [63]. Aerial imagery could not be found for the Hawkes Bay site; however, they are most likely at least this old as the trees are larger than the ones at Wairarapa. All pasture positions were between the slope gradients of 20° and 25° and were on the same northern hill slope.

The trees in each paddock were selected based on there being four or five individual kānuka trees and equivalent open pasture areas in close proximity of each other. Trees were selected in this way because soil moisture sensors were installed permanently into the soil and these had to be connected to a central data logger, and the cable lengths were no

more than 20 m long. Moreover, at both sites, there were livestock camping spots under a few trees on the downslope side, and these were specifically avoided when selecting study trees.

Measurements were taken for two years from 12 December 2019 to 11 December 2021. At each position, pasture production, soil fertility, soil physical properties, soil temperature, soil moisture and light interception were quantified.

Pasture production was measured using the pre-trimmed exclusion technique [64]. Pasture was harvested using one pasture cage per position (*n* = 9 per treatment). For each cage, pasture was pre-trimmed with electric clippers to 1 cm [65]. The pasture was harvested to 1 cm from a 25 cm × 50 cm quadrat within the pasture cage area after a ~2-month regrowth period (depending on the season and pasture height). Both sites were cut either on the same day or on consecutive days. After the quadrat area was sampled, the cage was moved to a new pre-trimmed pasture spot within the same position. Cages were rotated between three pasture cage spots within each position. This allowed livestock grazing and nutrient return to continue within the sampling locations throughout the study. If there were obvious dung or urine deposits where a cage was to be placed, an alternative position was used. After collection, each sample was oven-dried for 72 h at 70 °C and weighed. Every season, a subsample was taken and split into live and dead matter groups. These subsamples were also oven-dried for 72 h at 70 °C and weighed. This meant dead matter and green dry matter production (GDMP) (total weight minus dead matter) could be compared between treatments.

Volumetric soil moisture (VSM) was continuously measured using time domain reflectometry Campbell Scientific CS616 soil moisture sensors (sensor length 30 cm) (Campbell Scientific, Logan, UT, USA). Sensors were installed vertically at two depths (0 cm to 30 cm and 30 cm to 60 cm) in the center of each measurement position. There was one Campbell Scientific CR1000 data logger at the Hawkes Bay site with cables extended using two Campbell Scientific 16/32B multiplexers. One Campbell Scientific CR1000 data logger and one Campbell Scientific CR800 data logger were used to collect data from the Wairarapa site. All data loggers and multiplexers were contained in waterproof electrical boxes, connected to a 12 V battery that was charged by a solar panel. Data loggers took readings every 30 min. VSM measurements were averaged over the measurement period for each position (*n* = 9 per treatment), and VSM summer measurements were defined as VSM measurements between 15 December and 14 March.

Soil fertility was systematically sampled from a pasture cage spot in each position using ten soil cores (0–7.5 cm) in December 2019 and December 2021. After sampling, the cores were bulked together to form one representative sample per position. Soil samples were then sent to a testing laboratory (Hills Laboratories, Hamilton; Certified NZS/ISO/IEC 17025:2005 by International Accreditation New Zealand) where they were analyzed for pH (1:2 soil to water) [66], Olsen phosphorus (Olsen-P, 30 min bicarbonate extraction followed by Molybdenum Blue Colorimetry) [67], soil organic matter (Dumas combustion was used to calculate total carbon, and organic matter was 1.72 × total carbon) [68], total nitrogen (total-N, Dumas combustion) [68], sulfate sulfur (sulphate-S, 0.02 M potassium phosphate extraction followed by Ion Chromatography) [69], sodium/potassium/magnesium/calcium (Na/K/Mg/Ca, 1M neutral ammonium acetate extraction followed by ICP-OES) [66] and cation exchange capacity (CEC, summation of extractable cations (K, Ca, Mg, Na) and extractible acidity) [70]. Both the measurements from December 2019 and December 2021 were averaged to form a single soil fertility measurement for each position (*n* = 9 per treatment). Measurements were not repeated measures over time, but averaged over both years, because different cage spots were sampled in each position to avoid damaging the soil in a single cage spot.

The 3 cm (height) by 4.8 cm (diameter) soil cores were taken at 2–5 cm in the topsoil in September 2021 and used to measure bulk density, particle density, pore size distribution and the water retention curve. Four cores were sampled 50 cm to the left of the VSM sensor in each position and averaged to form one measurement per position (*n* = 9 per treatment).

Particle density was calculated using a subsample from one replicate per position according to the method described by Gradwell and Birrell [71], and with bulk density used to calculate porosity for the topsoil (porosity of 2–5 cm). For the pore size distribution and water retention curve, cores were saturated from below and then equilibrated at the matric potential values of −6 kPa (hanging water column) and −1500 kPa (in a pressure chamber), which correspond to the pore sizes of <54 μm and <0.2 μm, respectively. Macroporosity was defined as pore sizes draining between 0 kPa (saturation) and −6 kPa [72]. Plant available water capacity was defined as water draining between −6 kPa and −1500 kPa [72]. Bulk density was also measured between the depths of 40 cm and 45 cm using 5 cm (height) by 4 cm (diameter) soil cores taken using a soil corer. Bulk density was measured at this depth because it was within the depth of the soil moisture sensors between 30 cm and 60 cm.

Photosynthetically active radiation (PAR) was measured 50 cm above the pasture in the center of each position using a Skye Spectro Sensor 2 data logger attached to a Skye PAR sensor (Skye Instruments, Llandrindod Wells, UK). PAR was measured one day per season 30 times at solar noon, solar noon +2 h and solar noon −2 h on a cloudless day during the second year. Measuring over a 4 h window captured tree shading variation during the day. After each set of 30 measurements at each tree position, 1 measurement was taken in the paired open pasture position. Light interception (LI) by the kānuka trees was calculated by subtracting each kānuka pasture PAR measurement from the paired open pasture PAR measurement. One LI measurement was formed per kānuka pasture position by averaging all the measurement times and seasons (*n* = 9 per treatment).

Soil temperature was measured using temperature MicroLoggers (Hortplus, Hamilton, New Zealand) placed at a 5 cm depth 10 cm to left or right of the soil moisture sensor between mid-December 2019 and mid-August 2020. This spanned the first summer (from mid-December to mid-March) and winter (from mid-June to mid-August). The loggers measured temperature every 3 h beginning at 12:01 pm. The measurements at 12:01:00 and 15:01:00 were defined as the day-time temperatures, and the measurements at 00:01:00 and 03:01:00 were defined as the night-time temperatures. Although loggers were placed in all kānuka pasture positions, 5 units malfunctioned, so 5 loggers measured at Wairarapa (2 in kānuka pasture and 3 in open pasture) and 8 measured at Hawkes Bay (3 in kānuka pasture and 5 in open pasture).

At each site, an Onset Hobo RX3000 remote monitoring station (Onset Computer Corporation, Bourne, MA, USA) was installed in open pasture which recorded precipitation (mm), air temperature (°C), relative humidity (%) and wind speed (m s$^{-1}$). The rain gauge at Hawkes Bay malfunctioned during the study, so rainfall data for each year were used from a weather station 5.8 km from the site (station 2523) [60].

### 2.3. Statistical Analysis

Mixed-effect models were used to compare variables between treatments (kānuka pasture and open pasture) [73,74]. Treatment was a fixed effect and site was a random effect. Interactions between treatment and site were also calculated. GDMP, dead matter, pH, sulfate-S, porosity 2–5 cm, available water capacity 2–5 cm, macroporosity 2–5 cm, bulk density 40–45 cm, VSM 0–60 cm, VSM 30–60 cm, summer VSM 0–30 cm and summer VSM 30–60 cm were tested without transformation as model residuals were approximately normal and their variances homogeneous after visual assessment [73,74]. Green:dead matter ratio, pH, total-N, organic matter, CEC, K, Ca, Mg and Na were tested after being log-transformed so the model assumptions were met. Soil temperature was not statistically tested between treatments because of the reduced sample sizes after some of the sensors malfunctioned.

The multivariate canonical variate analysis (CVA) was then used to find the variables that best explained the variation between the treatments (kānuka pasture and open pasture) and sites (Wairarapa and Hawkes Bay), after the data were normalized [75,76]. Variables that respond in a similar way in terms of how they impact pasture production were not duplicated in the CVA analysis because this distorts the model, overestimating the

influence of the duplicated variables. Therefore, VSM 0–30 cm was used as the sole VSM measurement, and CEC was used instead of all the cations because it is the summation of K, Na, Mg and Na. K was kept in the analysis because it is often the most important cation for plant growth [77]. Only topsoil soil physical variables were used because these likely had more influence on pasture growth than bulk density 40–45 cm, and pH was not included in the model because pH was very similar between the treatments.

All the statistical analysis was performed on R (v.4.1.1) [78]. The 'lme4' package was used to make the mixed-effect model [79] and the 'candisc' package was used to do the CVA analysis and create the biplot [80].

## 3. Results

### 3.1. Pasture Production

On average between the two sites, there was 107.9% ($p < 0.001$) more GDMP in kānuka pasture than open pasture (Table 2). In year 1 and 2, there was 137.7% ($p < 0.001$) and 85.0% ($p < 0.001$) more GDMP in kānuka pasture, respectively. There was significantly more dead matter in the open pasture ($p < 0.001$). There was a significant interaction for green:dead matter ratio ($p < 0.01$) between the treatment and site (Figure 2).

**Table 2.** GDMP (green dry matter production), dead matter and green:dead matter ratio for the treatments averaged between sites. The standard error of the mean is given in brackets.

| Variable | Kānuka Pasture | Open Pasture | Significance |
|---|---|---|---|
| GDMP (kg ha$^{-1}$ yr$^{-1}$) | 5541.0 (747.8) a | 2665.8 (333.8) b | *** |
| Dead matter (kg ha$^{-1}$ yr$^{-1}$) | 681.8 (91.2) b | 1014.8 (123.7) a | *** |
| Green:dead matter ratio | 8.8 (1.3) a | 2.7 (0.2) b | *** |

Different letters represent significant differences between the treatments; *** = $p < 0.001$ level.

### 3.2. Factors Influencing Pasture Production

Rainfall in year 1 and 2 was 786 mm and 686 mm at the Wairarapa site, respectively, and 772 and 835 mm at a weather station 5.8 km from the Hawkes Bay site, respectively. The average temperature was 13.2 °C (min = −0.4 °C; max = 31.8 °C) and 14.1 °C (min = 0.3 °C; max = 37.1 °C) in year 1 and 2 at Wairarapa, respectively, and 12.5 °C (min = −2.5 °C; max = 34.6 °C) and 13.3 °C (min = 0.8 °C; max = 30.8 °C) at Hawkes Bay, respectively. The average windspeed at Wairarapa over two years was 1.3 m s$^{-1}$ (min = 0.0 m s$^{-1}$; max = 16.4 m s$^{-1}$) and 2.5 m s$^{-1}$ (min = 0.0 m s$^{-1}$; max = 16.0 m s$^{-1}$) at Hawkes Bay.

There was 67.2% LI under the kānuka trees at the Wairarapa site and 51.2% at the Hawkes Bay site. The minimum and maximum LI over the kānuka pasture positions at Wairarapa was 49.9% and 85.7%, respectively, and the minimum and maximum LI at Hawkes Bay was 23.5% and 88.0%, respectively.

The soil fertility variables Olsen-P ($p < 0.001$), K ($p < 0.001$), Mg ($p < 0.01$) and Na ($p < 0.001$) were significantly greater in the kānuka pasture (Table 3). There was a significant interaction for total-N ($p < 0.01$), organic matter ($p < 0.01$), CEC ($p < 0.01$) and Ca ($p < 0.001$) (Figure 2). Porosity 2–5 cm ($p < 0.05$) was also significantly greater in the kānuka pasture (Table 3).

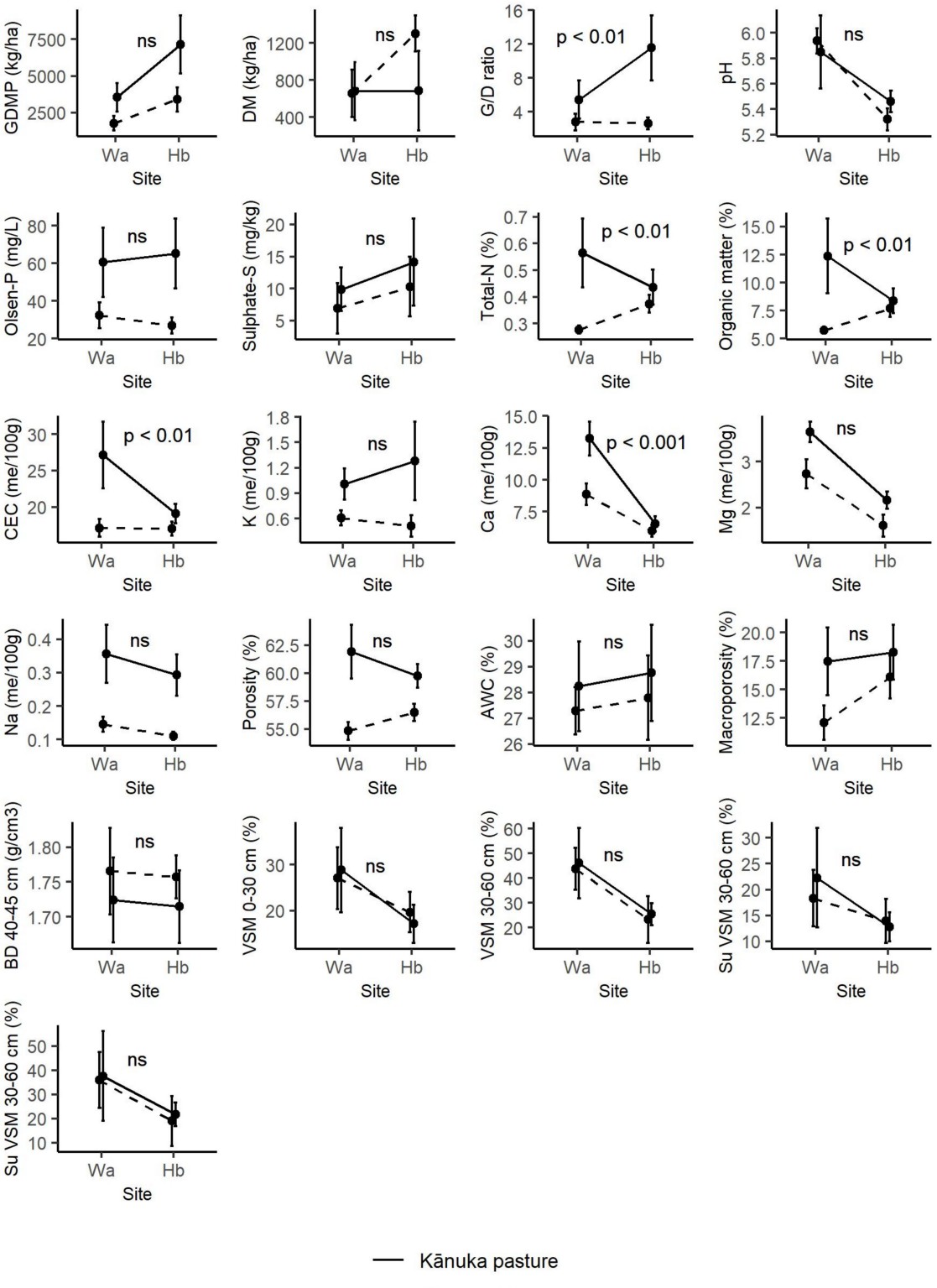

**Figure 2.** Pasture production and abiotic factor treatment and site interactions. The error bars are the 95% confidence intervals. ns = not significant. Wa = Wairarapa. Hb = Hawkes Bay. GDMP = green dry matter production. DM = dead matter. G/D = green:dead. P = phosphorus. S = sulphur. N = Nitrogen. CEC = cation exchange capacity. K = potassium. Ca = calcium. Mg = magnesium. Na = sodium. Porosity = porosity 2–5 cm. AWC = available water content 2–5 cm. Macroporosity = macroporosity 2–5 cm. BD = bulk density. VSM = volumetric soil moisture. Su = summer.

**Table 3.** Soil variable measurements for the treatments averaged between sites. The standard error of the mean is given in brackets.

| Variable | Kānuka Pasture | Open Pasture | Significance |
|---|---|---|---|
| pH | 5.6 (0.3) a | 5.6 (0.3) a | ns |
| Olsen-P (mg L$^{-1}$) | 63.2 (5.6) a | 29.3 (1.7) b | *** |
| Sulphate-S (mg kg$^{-1}$) | 12.1 (1.7) a | 8.8 (1.4) a | ns |
| Total-N (%) | 0.5 (0.03) a | 0.3 (0.01) a | ns |
| Organic matter (%) | 10.1 (0.8) a | 6.8 (0.3) a | ns |
| CEC (mg 100 g$^{-1}$) | 22.7 (1.3) a | 17.1 (0.3) a | ns |
| K (mg 100 g$^{-1}$) | 1.2 (0.1) a | 0.6 (0.04) b | *** |
| Ca (mg 100 g$^{-1}$) | 9.5 (0.1) a | 7.3 (0.04) a | ns |
| Mg (mg 100 g$^{-1}$) | 2.8 (0.2) a | 2.1 (0.2) b | ** |
| Na (mg 100 g$^{-1}$) | 0.3 (0.02) a | 0.1 (0.007) b | *** |
| Porosity 2–5 cm (%) | 60.6 (0.6) a | 55.7 (0.3) b | * |
| Available water capacity 2–5 cm (%) | 28.5 (0.6) a | 27.6 (0.4) a | ns |
| Macroporosity 2–5 cm (%) | 17.9 (0.8) a | 14.3 (0.7) a | ns |
| Bulk density 40–45 cm (g cm$^{-3}$) | 1.72 (0.02) a | 1.76 (0.02) a | ns |

Different letters represent significant differences between the treatments; * = $p < 0.05$; ** = $p < 0.01$; *** = $p < 0.001$ level; ns = not significant. P = phosphorus. S = sulfur. N = nitrogen. CEC = cation exchange capacity. K = potassium. Ca = Calcium. Mg = Magnesium. Na = sodium.

There were no significant differences between treatments for any VSM variables nor significant interactions (Table 4; Figure 2). At no point was the soil saturated at either site during the study as the VSM was never greater than the porosity at any position between 0 cm and 30 cm.

**Table 4.** Volumetric soil moisture (VSM) for the treatments averaged between sites. The standard error of the mean is given in brackets.

| Variable | Kānuka Pasture | Open Pasture | Significance |
|---|---|---|---|
| VSM 0–30 cm (%) | 23.0 (2.5) a | 22.3 (1.8) a | ns |
| Summer [a] VSM 0–30 cm (%) | 17.0 (6.4) a | 15.9 (3.9) a | ns |
| VSM 30–60 cm (%) | 33.2 (4.0) a | 33.5 (4.3) a | ns |
| Summer [a] VSM 30–60 cm (%) | 27.7 (9.6) a | 27.5 (11.1) a | ns |

Different letters represent significant differences between the treatments; ns = not significant; [a] Summer = VSM measurements between 15 December and 14 March.

The mean day soil temperature in the first summer at Wairarapa under and away from the trees was 20.6 °C (min = 14.8 °C; max = 26.4 °C) and 25.5 °C (min = 16.9 °C; max = 36.5 °C), respectively, and at Hawkes Bay it was 19.2 °C (min = 12.3 °C; max = 26.7 °C) and 21.9 °C (min = 13.5 °C; max = 30.5 °C), respectively. The mean night soil temperature in the first winter at Wairarapa under and away from the trees was 11.3 °C (min = 4.0 °C; max = 19.1 °C) and 12.4 °C (min = 4.5 °C; max = 26.4 °C), respectively, and at Hawkes Bay it was 11.1 °C (min = 4.5 °C; max = 19.1 °C) and 11.1 °C (min = 4.5 °C; max = 19.3 °C), respectively. The mean day soil temperature in the first winter at Wairarapa under and away from the trees was 13.2 °C (min = 6.6 °C; max = 22.9 °C) and 14.3 °C (min = 6.4 °C; max = 23.0 °C), respectively, and at Hawkes Bay it was 12.9 °C (min = 5.1 °C; max = 22.8 °C) and 13.2 °C (min = 5.0 °C; max = 23.0 °C), respectively.

### 3.3. Canonical Variate Analysis

The Canonical Variate Analysis (CVA) explained 92.2% of the total variation between the treatments and sites (Figure 3). The Wilks' lambda was significant ($p < 0.001$). Canonical variate 1 explained 72.3% of the variation ($p < 0.001$) and canonical variate 2 explained 19.9% ($p < 0.05$). Canonical variate 3 explained 7.8% of the variation ($p > 0.05$). The first canonical variate discriminated the data per treatment (kānuka pasture and open pasture) (x-axis) and canonical variate 2 discriminated the data per site (Wairarapa and Hawkes Bay) (y-axis). Olsen-P, K and porosity were the environmental variables most strongly

positively associated with kānuka pasture. VSM 0–30 cm was most strongly positively associated with the Wairarapa site. GDMP was most strongly positively associated with kānuka pasture at the Hawkes Bay site.

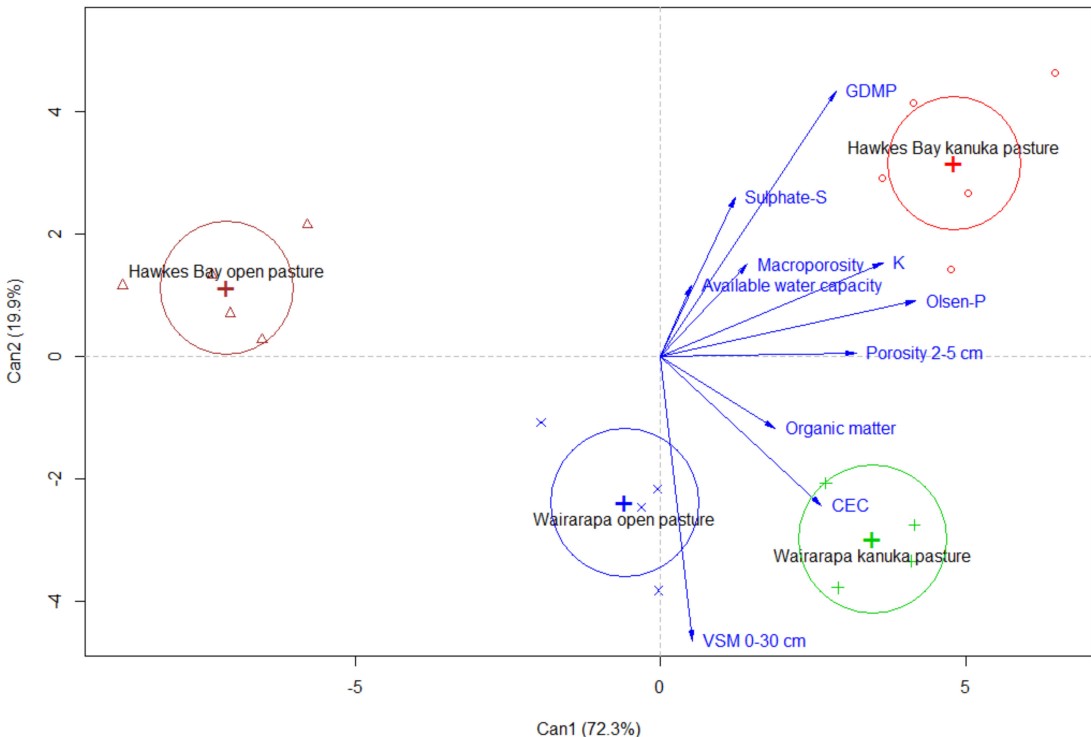

**Figure 3.** Canonical variate analysis showing which variables best explain treatment and site differences. GDMP = Green dry matter production. P = phosphorus. S = Sulfur. K = potassium. CEC = cation exchange capacity. VSM = volumetric soil moisture.

## 4. Discussion

The 107.9% greater green dry matter production (GDMP) under kānuka silvopastoral trees shows that the genus has potential to increase the production of low-producing sloped areas of New Zealand hill country. This result is contrary to past silvopastoral research in hill country, with no published studies finding increased pasture production under mature trees in hill country compared to equivalent areas of open pasture [24,81,82]. These results open the door for continued work in hill country on silvopastoral trees that have an overall facilitating relationship, as opposed to an overall competitive relationship, with pasture production.

### 4.1. Soil Nutrients

It is widely documented that silvopastoral trees can improve soil fertility under their canopies [2], with there being many examples from oak silvopastoral systems in southern Europe [1,3–5] and California [38,83,84]. Soil nutrient increases under trees measured in previous studies are comparable to the current study, with Dahlgren et al. [83] finding 55–60% higher organic carbon and N pools under an oak canopy in California. Moreover, Rossetti et al. [3] found over 50% higher organic matter and available P under oak canopies in Italy.

Olsen-P and K concentrations were very high under the kānuka trees in the current study compared to those under poplar silvopastoral trees [85,86], in past hill country research comparing medium and high sloped areas [65], and research established optimum levels for maximizing hill country pasture growth [87]. Moreover, these soil variables were strongly positively associated with kānuka pasture in the CVA. One likely mechanism con-

tributing to these greatly elevated nutrient levels is livestock depositing and concentrating urine and dung in the kānuka pasture environment [2].

This could be happening for two reasons. Open pasture at both sites were exposed to strong winds and sun, and the tree-pasture environment likely represented a sheltered and shaded environment. Secondly, pasture could have been preferentially grazed in kānuka pasture. López et al. [88] has previously provided evidence that perennial ryegrass is preferentially grazed in hill country, and the greater pasture production under the trees at both these sites in the current study has been shown to be the result of the growth of more productive pasture species, such as perennial ryegrass (Mackay-Smith et al. *under review*).

Livestock nutrient transfer for P and S by livestock from medium and high sloped hill country areas to low sloped areas has been shown by Saggar et al. [89], and nutrient transfer is one of the main reasons for the poorer soil conditions and reduced pasture growth in steeper areas of hill country [65,89]. Therefore, this study provides evidence that nutrient transfer by livestock could also potentially occur from open pasture areas to tree pasture areas, and trees might be able to be used as a tool for nutrient transfer and spatial distribution.

Another factor that could be contributing to a build-up of organic matter under the silvopastoral trees, in addition to livestock urine and dung deposition, is tree litterfall. Litterfall in mānuka-kānuka scrub (mānuka (*Leptospermum scoparium* J.R.Forst. & G.Forst.) is a tree that is in the same family as kānuka and both often grow together in mixed shrubland) has been shown to add 1941–2488 kg ha$^{-1}$ yr$^{-1}$ of carbon and 28–37 kg ha$^{-1}$ yr$^{-1}$ of N to the soil [90]. This study was in a 'high-density' (no specific density was defined in the study) unmanaged stand that also had forest undergrowth, so the system studied by Lambie and Dando [90] would have most likely added more litter than individually spaced kānuka trees in a silvopastoral system. Nevertheless, this offers evidence that kānuka trees should add organic matter and N to the soil.

The interaction between site and treatment for organic matter shows that there was over 50% more organic matter in kānuka pasture compared to open pasture at Wairarapa, but organic matter was similar between treatments at Hawkes Bay (Figure 3). This could be because organic matter had reached an optimum level in both kānuka pasture and open pasture at Hawkes Bay, but not at Wairarapa. This may also explain why GDMP was lower in the open pasture at Wairarapa when compared to the open pasture at Hawkes Bay.

Other reasons reported in the past literature for trees increasing the availability of nutrients under silvopastoral trees include nutrient enrichment of throughfall by the tree canopy [38,91,92], or the addition of mycorrhizal fungi in the soil facilitating nutrient uptake by pasture [93,94]. Although mycorrhiza activity has been shown to be negatively impacted by high soil P levels [95], so mycorrhiza activity may not be a significant factor at the studied sites.

### 4.2. Soil Water and Structure

In terms of soil water, trees can facilitate water availability improvements by adding shade and reducing wind run, which can reduce evapotranspiration and water loss from the soil [2,7,12]. Trees can also modify soil physical properties, which can lead to increased water retention [6], and hydraulically uplift water from lower soil layers [96,97]. Yet, trees also take up water, depleting available water resources for plant growth [31,32] and intercept rainfall, reducing the amount of rainfall reaching the soil [31]. Nevertheless, the VSM results offer evidence that the kānuka trees were not outcompeting pasture for VSM, but they were also not conserving VSM compared to open pasture. This result is positive because it shows that the tree water use, or rainfall interception, were not having overriding negative influences on the system and negatively impacting pasture production.

The improvements in porosity may have contributed to pasture production improvements directly, as improvements to porosity can increase the growth roots [98,99] and facilitate root aeration [77,100]. Because of the evidence that trees provide shelter to livestock under the trees, it is surprising that this did not result in soil compaction, because it

is well established that increased livestock activity can result in negative impacts to soil physical properties [101–103]. For example, Zhang et al. [103] found porosity increased from 0.64 to 0.78 when the grazing intensity was reduced from 4.8 animal unit months ha$^{-1}$ to 1.2 animal unit months ha$^{-1}$ in Canadian pastoral land. The soil physical results from the present study, thereforem indicate that the potential livestock activity under the trees was not intensive enough to result in negative impacts to soil structure.

### 4.3. Tree Bio-Physical Attributes

The conclusion in past poplar studies is that light was the main limiting factor to pasture growth in studies that have measured less pasture production under poplars in wetter and drier areas of sloped hill country [34,36,37]. The results of this study question this conclusion because, following that reasoning, pasture production should have also been less under the kānuka trees because there was on average 67.2% and 51.2% light interception by the trees at Wairarapa and Hawkes Bay, respectively. Therefore, this is evidence that the contrasting bio-physical attributes of poplars led to reduced pasture production reported in past studies [24] and not light reductions.

Past poplar studies have found that the trees do not build-up Olsen-P compared to open pasture [85,86]. The contrasting results in this study could be a result of differing livestock interactions under poplar and kānuka canopies. Kānuka is a much smaller tree than poplar and has a more sheltered environment [49]. As such, livestock may prefer to spend more time under kānuka trees compared to poplars, resulting in more P nutrient transfer to kānuka pasture positions [65,89]. Moreover, kānuka trees are evergreen and poplars are deciduous, which means kānukas could potentially facilitate more livestock use under tree canopies throughout the year.

Past research does not find conclusive evidence that poplar trees facilitate organic matter increases under tree canopies compared to open pasture [85,86]. This contrasts to the substantially greater organic matter percentages under the kānuka trees at Wairarapa, and the similarly high organic matter levels in both the kānuka pasture and open pasture at Hawkes Bay. It is especially surprising that some of the sites measured by Guevara-Escobar et al. [85] and Wall [86] found decreased organic matter under tree canopies because poplars also add litterfall to the soil annually [104]. A reason for this could be livestock grazing these leaves when they are on the ground, as poplar leaves are highly palatable to livestock when they are green and can be used as supplementary fodder [23,105].

There is evidence that poplars reduce soil moisture compared to open pasture, with Douglas et al. [34] reporting 33% reductions of soil moisture (0–20 cm soil depth) under poplars in a summer and autumn drying phase compared to open pasture. Moreover, Guevara-Escobar et al. [36] also found evidence of less soil moisture (0–15 cm soil depth) in late summer (March) and autumn (May) under poplar trees compared to open pasture. The similar VSM measurements in summer in this present study are encouraging because it offers evidence that kānukas are potentially not depleting soil moisture in summer compared to open pasture.

Finally, poplars may have directly negatively impacted pasture production through their leaf fall smothering grass in autumn [26,104]. As the leaf fall of kānukas is spread throughout the year [90], litterfall should potentially have less of a negative impact on pasture production in a kānuka silvopastoral system.

A facet not investigated in this study is the influence of silvopastoral trees on pasture production in different climates. Rivest et al. [106] provides evidence that the relationship between the impact of silvopastoral trees on pasture production and annual precipitation depends on tree type. The authors found a negative linear relationship between effect size on pasture production and annual average precipitation for N-fixing silvopastoral trees, but a positive linear relationship for eucalyptus (*Eucalyptus* spp.) [106]. This highlights how trees with different attributes can result in contrasting pasture production outcomes. The impact of kānuka on pasture production could therefore vary depending on rainfall, and thus have a different climate–production relationship compared to poplars.

*4.4. Limitations*

It is important to recognize that this study is the first study to measure the influence of kānuka silvopastoral trees on soil properties and pasture production. New Zealand pastoral land is a highly variable landscape with contrasting soil types, climates, topographies, aspects, management types and livestock types. The impact of kānukas on pasture will most likely vary with these conditions. More work is required on other farms in these different conditions to form generalized conclusions for how kānuka performs as a silvopastoral tree in hill country.

Another caveat is that livestock camping areas were specifically avoided in tree selection in the current study due to their tendency to denude pasture. These livestock camping areas are likely a result of the silvopastoral tree design and livestock management. More work is required to understand the dynamics of livestock management and camping areas in hill country, and how they might impact the overall potential positive impacts of kānukas on pasture production at farm scale.

Furthermore, trees were selected at each site based on their close proximity to other individually spaced kānuka trees and to match equivalent areas of open pasture because of equipment constraints. It is possible trees growing closer together may have resulted in different livestock interactions or tree influences compared to more isolated silvopastoral trees. Nevertheless, from the visual observation of the silvopastoral environments at both sites, the environment studied represented the typical agricultural environment under the kānuka trees that did not have livestock camping spots.

**5. Conclusions**

Over a two-year period at two sites, this study measured how a novel silvopastoral system in New Zealand with kānuka, which has similar bio-physical attributes to the ñire forests of southern Patagonia and the oak silvopastoral systems of southern Europe and California, influences pasture production and pasture–soil relationships. There was 107.9% increase in pasture production in kānuka pasture positions, and Olsen-P, porosity and K best explained the variation between kānuka pasture and open pasture positions. Volumetric soil moisture was statistically similar in kānuka pasture and open pasture positions. The high concentration of soil P and K in kānuka pasture offers evidence of nutrient transfer via grazing stock to the tree-pasture environment. Moreover, there was 48.6% more organic matter under the trees and a significantly greater porosity, which is evidence that other processes also contribute to soil organic matter levels in the kānuka pasture environment, such as litterfall.

These results are evidence that kānuka has great potential as a silvopastoral tree for forming transformative multifunctional landscapes, potentially adding both environmental and economic value to New Zealand hill country farms.

**Author Contributions:** Conceptualization, T.H.M.-S., I.F.L., L.L.B., J.I.R.; Methodology, T.H.M.-S., I.F.L., L.L.B.; Formal Analysis, T.H.M.-S.; Data Curation, T.H.M.-S.; Writing—Original Draft Preparation, T.H.M.-S.; Writing—Review and Editing, I.F.L., L.L.B., J.I.R.; Supervision, I.F.L., L.L.B., J.I.R.; Project Administration, L.L.B.; Funding Acquisition, T.H.M.-S. All authors have read and agreed to the published version of the manuscript.

**Funding:** This work was supported by the Massey University doctoral scholarship program, New Zealand; Greater Wellington Regional Council, Masterton, New Zealand (contract RM22793); the New Zealand Ministry of Business, Innovation and Employment research program "Smarter Targeting of Erosion Control (STEC)" (contract C09X1804); and the C. Alma Baker Trust, New Zealand (contract RM23081).

**Institutional Review Board Statement:** Not applicable.

**Informed Consent Statement:** Not applicable.

**Data Availability Statement:** The data presented in this study are openly available in Zenodo at https://doi.org/10.5281/zenodo.

**Acknowledgments:** The authors would like to thank Mark Guscott and Jeremey Rookes for very kindly letting us use their farms; Bob Toes and Ross Gray for their help in installing the weather stations; Alan Palmer for describing the soils at each site; and Dougall Gordan (Greater Wellington Regional Council), Petra Fransen (Greater Wellington Regional Council) and Chris Phillips (Landcare Research) for providing research funding to the project; and Martin Smith for proofreading the document.

**Conflicts of Interest:** The authors declare no conflict of interest.

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
