# Peer review of "Kānuka Trees Facilitate Pasture Production Increases in New Zealand Hill Country"

_agronomy, doi:10.3390/agronomy12071701_

Round 1

Reviewer 1 Report

General Comments:

The manuscript reviewed was “A fresh approach to silvopastoralism in New Zealand with kanuka”. The study evaluated biomass production and fertility in shaded and unshaded areas near the kanuka tree in the pasture. There are a few grammatical errors, but overall, the manuscript is well written. The work has technical merit, but the system seems to be categorized inappropriately: is this really silvopasture?

The abstract, intro and discussion lack direction and purpose and seems to contain unnecessary information. The results are not well discussed or supported by other literature. I would stay away from phrases such as “trees not in New Zealand most likely grow slower than poplar” – if you write a statement, support it with literature – if you cannot, then don’t write that statement. I appreciate that there may not be much information available about this topic and kanuka specifically but stay away from speculation. I think the lack of available information might also be influencing the author to make comparisons to the poplar tree and spend a considerable amount of time describing the poplar tree – but this paper is about kanuka – so talk more about kanuka, the manuscript does not describe kanuka well – also poplar was not in this study, so the authors are limited on the comparisons between poplar and kanuka.

The study also lacks important forage information that is required to make the claim that this is a fresh approach to silvopastoralism. Forage is a huge component of a silvopasture, yet the only information provided is total biomass. What is the botanical composition? Two species are mentioned as typical forages for the area – perennial ryegrass and browntop. There’s no description of these forages or their relationship to shade or marginal fertility – as I understand perennial ryegrass requires high fertility while browntop is in marginal areas? – Is more ryegrass found in high fertility areas under the shade while browntop might be in the low fertility/unshaded areas?

I think the conclusions here are oversold. I wouldn’t consider this a silvopasture systems study – the study just evaluates fertility near the tree and some distance from the tree. There is no other information related to silvopasture. By most silvopasture definitions, this would also not be considered silvopasture because the tree is not providing a commercial product - timber/fruit/nuts/etc.. and these products an intentional part of the system – can shade be considered the tree product and is this silvopasture? The title and abstract are misleading because I was expecting an actual silvopasture study with replicated pastures, livestock, forage measurements, etc…

Having said that…I think the information is still great. I like the concept that trees could be used to influence livestock behavior on the landscape by encouraging them to loaf around trees located in low fertility areas (side slopes) and improve fertility and subsequent forage production and livestock gain/performance and overall production for that land. If this is what you mean by a fresh approach to silvopastoralism – I might buy it, but it was not clear to me in the text that that was the meaning of a fresh approach. You may need to better describe the previous/old approach and show the reader how they differ – perhaps that was what you were doing with the poplar information – but it really was not clear to me as a reader, I think some rearranging and more intentional messaging (removal of unnecessary information) will help the reader connect the dots.

Abstract:

The abstract is not very helpful. It does not tell me the experimental design. Without reading the whole manuscript I don’t know what “positions” means. For the results, part of it, includes the data and p values.

Introduction:

I would start this intro with the description of NZ hill pastures – describe the problem. Then describe the problem with “islands of fertility” and then connect the ideas – islands of fertility could solve fertility issues on the hill pastures. I think this section has a lot of unnecessary information and that’s why your message/purpose is lost. The authors kind of dip into a discussion about native versus introduced species, and I would stay away from that, I am not sure its necessary unless you think it is really important in relation to how native trees interact with fertility versus introduced species…but since there are not introduced species in the experiment, this is all speculation and it can’t be supported with the data from the study anyway. More information about the forage species and their interactions with shade – performance in shaded and unshaded environments should be added. Maybe some notes about fertility would be nice too – maybe just a general comparison between high fertility and marginal areas in relation to forages and trees.

Methods: I think these are mostly well described. I could use some clarification on what the experimental unit is and what its size is. I am also interested in an explanation of why all the soil samples were composited over 2 years and why you did not include the year in the model. If the year was not included because it didn’t have many effects, that’s fine but state that – though, I would think the year would have a large effect on biomass production.

Results:

The big issue here is not having botanical composition for the forage data. Otherwise, I think the results are ok – some grammatical errors/wording need improvement. The first paragraph in the section Factors influencing pasture production might be better in a table.

Discussion:

Stay on point. The first sentence of the discussion is great, it really sums it up. But then, the point gets lost later in the paragraph with “This study gives evidence that kanuka may function differently to poplar as a silvopasture tree and that trees in hill country can significantly increase pasture production under the canopy” – so now you lost me again – you didn’t have poplar in the study. And then “silvopastoral trees that have a facilitating relationship, as opposed to a competitive relationship, with pasture production”. The forage production in that area is higher because of higher fertility – indirectly from the tree - it is a result of higher fertility. You are leaving out pieces of the puzzle. And trees and grasses are always competitive – just the benefits from fertility outweigh the detriments from shade (depending on the forage species – which we don’t know).

I think the discussion could include the nutrient requirements for the forage species and whether those nutrient requirements were met or different between shaded/unshaded environments. Also related to competitiveness – was rainfall/moisture limited in these 2 years? Do you know for sure they won’t compete under different environmental conditions?

It was mentioned in the intro and in the discussion, but I am not understanding the importance of tree age in relation to soil.

The last two paragraphs of the discussion seem unnecessary to me.

Wider considerations: I am not sure the purpose of this separately. Can this be integrated into other discussion sections?

Conclusion: This experiment is not an evaluation of a silvopasture system – the work just looks at fertility around the tree – this is not systems work. The second paragraph is unnecessary – no poplar in the study.

Reviewer 2 Report

Dear Authors,

The manuscript is written at a very high professional level, is adequately divided into individual chapters, the abstract is concise, the methodology is described accordingly, and the data are properly statistically processed, clearly presented, and adequately commented on in the discussion. The conclusions of the study were described correctly and are based on the results obtained earlier.

The manuscript is well written and its contents are appropriate.

My minor comment concerns the clarification in the table captions. The authors should make it clear in the captions to tables (2, 3, 4) that these are averages from two sites, as they explained at the beginning of row 249.

Congratulations to the authors of very interesting research (which I enjoyed reading), I wish them further success.
